# Motility Dynamics of T Cells in Tumor-Draining Lymph Nodes: A Rational Indicator of Antitumor Response and Immune Checkpoint Blockade

**DOI:** 10.3390/cancers13184616

**Published:** 2021-09-15

**Authors:** Yasuhiro Kanda, Taku Okazaki, Tomoya Katakai

**Affiliations:** 1Department of Immunology, Niigata University Graduate School of Medical and Dental Sciences, Niigata 950-8510, Japan; kandaya@med.niigata-u.ac.jp; 2Laboratory of Molecular Immunology, Institute for Quantitative Biosciences, The University of Tokyo, Tokyo 113-0032, Japan; tokazaki@iqb.u-tokyo.ac.jp

**Keywords:** T cell motility, tumor-draining lymph node, immune checkpoint blockade, CTLA-4, PD-1

## Abstract

**Simple Summary:**

Immune checkpoint blockade (ICB) therapies are attracting much attention for the clinical treatment of tumors. Combination therapies are being developed to enhance the effects of ICBs, and the importance of tumor-draining lymph nodes (TDLNs) has been reevaluated in antitumor responses via ICB therapy. The migration and motility status of T cells and their functional interaction with antigen-bearing dendritic cells are key factors for inducing adaptive immunity in LNs. Although immune checkpoint molecules are known to regulate T cell motility, their actual influence on T cell dynamics in TDLNs, particularly in ICBs, is poorly understood. In this review, we summarize the relevance of T cell dynamics and immune responses in draining LNs, and discuss the alteration of T cell motility under ICB. To develop better immunotherapies using ICBs, studying T cell dynamics in TDLNs can provide a good indicator to evaluate the efficacy of antitumor therapy.

**Abstract:**

The migration status of T cells within the densely packed tissue environment of lymph nodes reflects the ongoing activation state of adaptive immune responses. Upon encountering antigen-presenting dendritic cells, actively migrating T cells that are specific to cognate antigens slow down and are eventually arrested on dendritic cells to form immunological synapses. This dynamic transition of T cell motility is a fundamental strategy for the efficient scanning of antigens, followed by obtaining the adequate activation signals. After receiving antigenic stimuli, T cells begin to proliferate, and the expression of immunoregulatory receptors (such as CTLA-4 and PD-1) is induced on their surface. Recent findings have revealed that these ‘immune checkpoint’ molecules control the activation as well as motility of T cells in various situations. Therefore, the outcome of tumor immunotherapy using checkpoint inhibitors is assumed to be closely related to the alteration of T cell motility, particularly in tumor-draining lymph nodes (TDLNs). In this review, we discuss the migration dynamics of T cells during their activation in TDLNs, and the roles of checkpoint molecules in T cell motility, to provide some insight into the effect of tumor immunotherapy via checkpoint blockade, in terms of T cell dynamics and the importance of TDLNs.

## 1. Introduction

Lymph nodes (LNs) are a type of secondary lymphoid organ (SLO), located at strategic positions of the lymphatic vascular system, to filter the lymph fluid drained from peripheral tissues. Lymphocytes accumulate in LNs for the surveillance of antigenic information in lymph exudate, which triggers adaptive immunity [1]. Immune responses in LNs are initiated by the arrival of lymph-borne soluble or particulate antigens and/or activated dendritic cells (DCs), which capture antigens in the peripheral tissues through afferent lymphatic vessels [2]. For this purpose, LNs have a sophisticated structure to integrate the filtering/collecting antigens and to support/regulate dynamic immune cell activities.

The LN parenchyma comprises three major compartments—the follicles, paracortex, and medulla—each of which is constructed with an elaborate tissue and cellular microarchitecture [3]. The B and T lymphocytes circulating in the blood enter the LN via high endothelial venules (HEVs) in the paracortex, and further migrate into the follicles and paracortex, respectively [4]. A three-dimensional stromal meshwork supports the entire LN structure. The network of fibroblastic reticular cells (FRCs), which are mesenchymal stromal cells, provides a porous sponge-like structural backbone to the tissues [3,5]. FRC subsets with distinct functions are localized in each compartment; each subset attracts a set of hematopoietic cells by secreting unique chemokines and cytokines [6]. In the paracortex, the FRC meshwork provides a scaffold for T cell migration [4,5,7]. The space in the meshwork structure is approximately 10–20 µm wide, wherein T cells can move randomly in continuous contact with FRCs and DCs [3,8,9,10,11].

Tissue-derived soluble and particulate antigens in lymphatic fluid flow into the subcapsular sinus (SCS) of the LN, just below the capsule, via afferent lymphatic vessels. Specialized macrophages, and resident DCs in the SCS, capture antigens and transport them to the cortex and paracortex [12]. Furthermore, migratory DCs carrying antigens from the surrounding tissues also arrive at the SCS, and further migrate into the paracortex for antigen presentation to the T cells [13]. Live imaging of LNs, using two-photon microscopy, has revealed that naive T cells move around at high speeds (>10 µm/min) within the tissue environment of the paracortex [14,15,16,17]. Such active T cell motility is considered important for scanning the cognate antigens presented by DCs. From this viewpoint, changes in the migration modality of T cells in LNs are closely correlated with the efficiency and activation status of T cells through their interaction with DCs.

Likewise, adaptive immune responses against various tumors are likely initiated in tumor-draining LNs (TDLNs) (Figure 1). Tumor-derived soluble and particulate antigens entering the SCS through the afferent lymphatic route are captured by macrophages and DCs [18,19]. Alternatively, migratory DCs that have internalized antigens from tumor tissue migrate to the paracortex of TDLNs to prime antigen-specific T cells [20,21]. In particular, the cross-presentation of tumor antigens, internalized by DCs in the form of peptide-MHC class I complexes, is the key to prime CD8^+^ T cells and antitumor immunity [22,23]. The activated cytotoxic T cells exit TDLNs and arrive at the target tumor via blood circulation, to damage the tumor cells expressing cognate antigens. However, the mechanisms by which tumor cells evade host immune surveillance and, consequently, develop clinically emerging tumors remain largely unclear. One possible mechanism is the induction of an immunosuppressive environment in the primary tumor and TDLNs using inhibitory mediators. Currently, the most promising candidates in this regard are the so-called ‘immune checkpoint’ molecules, such as CTLA-4 and PD-1/PD-L1 [24,25].

In recent years, tumor immunotherapies that target immune checkpoint molecules using antibody-based inhibitors have been drawing increasing attention in clinical treatment [26]. The basic strategy is to trigger strong lymphocyte activation by blocking the function of the checkpoint molecules that negatively regulate immune responses. Importantly, signals transmitted by immune checkpoint molecules are also known to modulate T cell migration and motility, which can naturally be altered using checkpoint blockade [27,28,29].

In this review, we focus on the role of T cell migration/motility and TDLNs in antitumor responses. We, first, overview the transport and presentation of tumor antigens via DCs, and the correlation between interstitial motility and the priming/activation of antigen-specific T cells. Next, we discuss the relevant changes in T cell dynamics that are induced by the inhibition of immune checkpoint molecules for future perspectives.

## 2. Antigen Transport from Primary Tumor to TDLNs via Lymphatic Vessels

To exert antitumor responses, CD4^+^ and CD8^+^ T cells that are specific to tumor antigens must receive adequate stimuli through the T cell receptor (TCR) via intimate interaction with DCs that display the antigenic peptide–MHC complex in the priming phase and subsequent effector phases [30,31]. The selective enhancement of antitumor T cell responses in TDLNs is a promising strategy for tumor immunotherapy. In this regard, it is extremely important to understand the detailed processes by which DCs capture antigens in primary tumors and migrate toward TDLNs through the lymphatic vessels, prior to T cell priming.

Conventional DCs (cDCs) are the chief antigen-presenting cells (APCs) that engulf tumor antigens and transport them to draining LNs (Figure 1A). In mice, circulating cDC precursors (pre-cDCs) expressing the chemokine receptors CCR1 and CCR5 migrate into the tumor site through attraction to CCL3, which is produced by tumor cells [32]. Pre-cDCs then proliferate and differentiate into CD103^+^ DCs (cDC1) or CD11b^+^ DCs (cDC2) in the tumor tissue. cDC1s prime CD8^+^ T cells for antigen by cross-presentation, while cDC2s are specialized for priming CD4^+^ T cells. After internalizing the tumor antigens, these cDC subsets transmigrate into the lymphatic vessels via CCL21, CXCL12, and CX3CL1 [33]. Approximately one-third of cDC1 in tumor tissues express high levels of CCR7 (a receptor of CCL21, which is produced by lymphatic endothelial cells and LN stromal cells [20]). As the expression of lysosomal degradative enzymes in cDC1 is low, intact tumor antigens are retained in cDC1 (rather than in cDC2), and are transported to the LNs [21]. The migratory cDC1 arriving at LNs further moves into the paracortical T zone, and transfers tumor antigens to the LN resident CD8^+^ cDC1 subset [20,34,35]. For the cross-presentation of tumor antigens to naive CD8^+^ T cells in TDLNs, migratory cDC1 can be induced in the relatively early stages of tumorigenesis. In contrast, resident CD8^+^ cDC1 cross-prime tumor-specific T cells in the later stages, when they acquire a large number of antigens from migratory cDC1 [21,34]. There are several predicted mechanisms of antigen transfer between DCs (including exosomal transfer), although the most promising process is the synaptic transfer of vesicles via contact between DCs [34]. As in viral infections, the activation of CD4^+^ T cells by cDC2 is assumed to be required for licensing cDC1 to activate CD8^+^ T cells in antitumor immunity [36]. However, it has recently been clarified that cDC1 is responsible for priming CD4^+^ T cells, as well as CD8^+^ T cells [30,37].

## 3. Interstitial T Cell Migration in LNs

To obtain antigenic stimuli, naive T cells require direct contact with the DCs in the LNs [38]. The active migration of T cells within a confined tissue environment (interstitial migration) is a rational adaptation to increase the probability of encountering antigen-specific T cells with a limited number of antigen-presenting DCs. The rapid and random T cell movement, rather than directional motility, is thought to further increase the chance of identifying cognate antigens. The average velocity of T cell migration in the paracortex is approximately 10–15 µm/min [14,15,16,17]. The migration speed of lymphocytes within a tissue depends on the cell’s intrinsically defined actin polymerization/depolymerization rate [27], their adhesiveness to the surrounding substrates [39], the extrinsically defined migration cues mediated by chemokines and other chemical mediators [40,41], and the obstacles to the microenvironment (created by stromal cells, dendritic cells, and other lymphocytes) [17]. The FRCs in the paracortex construct a relatively regular meshwork, in which several DCs settle to form another dense network, thereby allowing migrating T cells to maintain continuous contact with either FRCs or DCs [5,7].

A gradient of chemokine concentration is known to induce directional cell movement (termed ‘chemotaxis’), whereas a uniform concentration supports non-directional and random motility (termed ‘chemokinesis’) [42]. The FRCs in the paracortex produce large amounts of CCR7 ligands, CCL19 and CCL21 [43,44], generating an environment with relatively uniform concentrations of these chemokines. Therefore, T cells expressing CCR7 exhibit random migration in this region. T cell motility is markedly reduced in the LNs of mice with *plt* mutations that lack both *Ccl19* and *Ccl21a* genes, indicating that these chemokines contribute to the active migration of T cells in the paracortex [45,46].

The integrin LFA-1 on T cells is expected to be required for active migration in the tissue environment. However, live imaging of LNs revealed that the actual contribution of LFA-1 in T cell migration within the LN parenchyma was estimated to be minimal, as the deficiency of LFA-1 and some integrin-associated processes showed only minor influences on T cell motility [41]. However, observations from LN tissue slicing, accompanied with acute functional inhibition, suggest that the high-speed fraction of T cell motility involves a low-affinity interaction between LFA-1 on T cells and ICAM-1 on DCs [39].

The lipid mediator lysophosphatidic acid (LPA) is also known to facilitate the chemokinetic motility of T cells in LNs. FRCs and HEV endothelial cells highly express the ectoenzyme autotaxin (ATX), which catalyzes LPA production. The antagonistic compounds of ATX enzyme activity or LPA receptors are reported to reduce T cell motility in LN slices [40]. Six different G-protein-coupled receptors (LPA1–LPA6) have been identified as receptors for LPA [47,48,49,50,51,52]. Among these, LPA2 expressed by T cells mediates interstitial motility, stimulating cell movement in a confined tissue environment [53,54].

## 4. Antigen Recognition by T–DC Interaction and Changes in T Cell Motility

Although DCs are less motile, particularly in the paracortex, their extending dendrites (which occupy two-thirds of the cell volume) demonstrate fast movements and enable the probing of approximately 5000 T cells per hour as they pass through the vicinity [55]. In contrast, naive T cells randomly move in the paracortex, along the network of FRCs, after exiting the HEVs [7] (Figure 1B). When naive T cells contact DCs presenting cognate antigens in the context of the peptide–MHC (pMHC) complex, they receive antigenic stimuli through the T cell receptors (TCRs) and dramatically change their motility. The sensitization of naive T cells requires the formation of an organized intercellular molecular assembly (known as the immunological synapse (IS)) with the interface of DCs [56]. In LNs, there are three distinct phases of T cell dynamics upon antigen recognition [57]. During the first scanning phase (phase 1), migrating T cells make brief intermittent contact with antigen-presenting DCs, and gradually accumulate the intracellular active forms of NFAT and c-fos [58,59]. In the next priming phase (phase 2), T cells are arrested on DCs in a stable and prolonged manner, through the formation of IS. During this phase, T cells begin producing IL-2 and IFN-γ, as well as increasing the expression of activation markers such as CD25, CD44, and CD69 [60]. The T–DC interaction lasts from a few minutes to several hours, but requires at least 6 h to gain T cell proliferation activity [61]. In the final phase (phase 3), T cells resume migration, away from DCs, and proliferate several times, followed by exiting LN as activated/effector cells. Some of the T cells remaining in the LN differentiate into central memory cells, and relocate to the periphery of the paracortex via the CXCL9/CXCR3 axis [62].

## 5. Immunological Synapse Formation and Stop Signal

The IS formed between the interface of the T–DC contact is a functional adhesive structure through which activation signals are transmitted into T cells. In vitro examinations of IS formation indicate that TCR/CD3 microclusters are rapidly formed within tens of seconds after initial contact, and, 5 min later, 100–200 microclusters are assembled at the center of the adhesion surface to form the central supramolecular activation complex (cSMAC) [63,64]. In the periphery of SMAC (pSMAC), adhesion molecules (LFA-1 on T cells and ICAM-1 on APCs) are accumulated to construct a tight adhesive belt surrounding the cSMAC. During this process, TCR–pMHC ligation transmits the ‘stop signal’ that leads to the reduction of T cell motility by changing the intracellular machinery and inducing strong integrin-mediated adhesion [65,66] (Figure 2A).

In general, TCR signaling alone is insufficient for T cell activation, and the additional signals transmitted by the ‘costimulatory’ receptors in the IS play a pivotal role. A typical costimulatory receptor expressed on naive T cells is CD28, which binds to CD80 and CD86 on APCs [67]. CD28 molecules accumulate in the outer margin of cSMAC, and their cytoplasmic part is associated with phosphatidyl inositol 3-kinase (PI3K) and protein kinase C θ (PKCθ), which activates the NF-кB and JNK/p38 signaling pathways. Based on examinations using the anti-CD28 antibody and CD28^−/−^ mice, it was revealed that CD28 is not involved in T cell motility and contact duration in DCs [68,69]. This suggests that the stop signal mainly depends on antigen-specific TCR ligation. On the other hand, inducible T-cell costimulator (ICOS), which is a CD28 family costimulatory molecule, is implicated in T cell motility. T cells in the border of the paracortex and follicle in LNs are triggered by ICOS ligand (ICOSL) to promote their motility, and are recruited into the follicles [70]. On the contrary, ICOS inhibits the motility of DCs and tumor cells through ICOSL ligation [71,72]. However, ICOSL can bind osteopontin and promote the mobility of tumor cells [73].

The ligation of immunoregulatory molecules, such as CTLA-4 and PD-1, which are close relatives of costimulatory receptors, potentially antagonizes the stop signals of T cells by suppressing TCR ligation and IS formation [28,29,69,74,75,76]. This ‘reversal of migration arrest’ prevents stable T–DC interaction and maintains T cell mobility.

## 6. Inhibition of T Cell Activation by CTLA-4 and PD-1

CTLA-4 and PD-1 are immunoglobulin superfamily transmembrane molecules that are known to negatively regulate T-cell activation.

CTLA-4 is a structural homolog of CD28, and a typical inhibitory receptor that is expressed on activated T cells and regulatory T cells (Tregs) [77,78]. In naive T cells, CTLA-4 does not participate in the initial T–DC interaction because it resides in the cytoplasmic vesicles, and its cell surface expression is induced 24 h after TCR stimulation [79]. Although the mechanistic details of CTLA-4-mediated inhibitory function have not been strictly determined, it is likely to cause antagonistic interference in CD28–CD80/CD86 interaction, due to the higher affinity or avidity of CTLA-4 binding to CD80/CD86, compared with that of CD28 [80]. In addition, CTLA-4 has an intracellular tyrosine motif (YVKM) that resembles the immunoreceptor tyrosine-based inhibitory motif (ITIM) in other inhibitory coreceptors. The association of SH2-domain-containing phosphatase-1 (SHP-1)/SHP-2 and protein phosphatase 2A (PP2A) with this motif leads to PI3K dephosphorylation and the inhibition of ZAP70 microcluster formation [81,82,83,84,85]. However, this intracellular process is still controversial, as some reports suggest that SHP-1/2 binding to the YVKM motif is unnecessary for CTLA-4-mediated suppression [83,86,87,88,89,90]. CTLA-4 also prevents T cell activation through cell-non-autonomous mechanisms; Tregs indirectly inhibit CD28 signaling in naive T cells by depleting CD80/CD86 on APCs through CTLA-4-mediated trogocytosis [91,92]. In LNs, this process selectively downregulates CD80/86 molecules in migratory DCs, rather than resident DCs [91].

PD-1 was identified as a functionally unknown molecule induced in hematopoietic cells upon apoptotic stimuli, and was later shown to be expressed on lymphocytes after antigenic stimulation [93,94]. When PD-1-deficient mice developed spontaneous autoimmune diseases, PD-1 was recognized as an inhibitory receptor [95,96]. PD-1 expression is also detected in exhausted T cells and Tregs [97,98,99,100]. PD-L1 and PD-L2 have been identified as ligands of PD-1 [101,102]. PD-L1 expression is detected in various cells, including immune cells such as DCs, lymphocytes, and macrophages, as well as in several organs, such as the heart, lung, liver, and pelvis, whereas PD-L2 is detected in DCs, macrophages, and mast cells, and only weakly in some organs [103,104]. PD-1 ligands are also induced in tumors or virus-infected cells, and their expression is enhanced by IFN-γ [105]. PD-1 has two intracellular tyrosine motifs, the immunoreceptor-tyrosine-switch motif (ITSM) and ITIM. The inhibitory function of PD-1 is dependent on ITSM phosphorylation, which preferentially recruits SHP-2 [106,107,108]. The ITSM motif serves a docking site for both SHP-1 and SHP-2 in vitro; however, only SHP-2 has been shown to interact with PD-1 in live cells [109]. SHP-2 dephosphorylates CD3ζ and its downstream signaling molecules [100,110]. One major difference in the intracellular processes of PD-1 and CTLA-4 is the mode of Akt inhibition; PD-1 ligation prevents the CD28-mediated activation of PI3K, whereas CTLA-4 signaling dephosphorylates Akt via PP2A [111]. Furthermore, it is reported that *cis*-interaction of PD-L1 with CD80 on the same APC prevents inhibitory signaling in T cells [112,113]. There are two possible mechanisms for the disruption of inhibitory signaling by *cis*-PD-L1–CD80 interaction. One is the partial overlap of the interface of PD-L1 to CD80 and PD-1 [112]. The other is disrupting the homodimerization of CD80 in APCs, which reduces the avidity of CTLA-4–CD80 interaction [113]. However, the *cis*-interaction does not inhibit the CD28 costimulatory signal, which allows PD-L1-expressing APCs to activate T cells.

In addition to naive T cells and effector/memory T cells, PD-1 is integrated into TCR microclusters and recruits SHP-2 in the IS at around 6 h after TCR stimulation [109,114]. T cells that receive PD-1 ligation are likely unable to maintain IS and stop signals, thereby passing through antigen-presenting DCs without stable arrest [60].

## 7. Immune Checkpoint Blockade and Activation of Antitumor Responses

At present, CTLA-4 and PD-1 are referred to as ‘immune checkpoint’ molecules. Immunotherapies using antibodies to inhibit their functions (termed ‘immune checkpoint blockade (ICB)’) have been developed to enhance antitumor responses. These ICB therapies are also likely to alter T cell dynamics through the modulation of IS formation and activation states in TDLNs, as well as in primary tumors [60].

Anti-CTLA-4 antibodies are conceptually thought to enhance the priming of naive T cells or sustained activation of memory T cells by interfering with the immunosuppressive CTLA-4–CD80/86 interaction [115,116]. However, this generally accepted view has not been rigorously proven [117]. In tumor eradication, anti-CTLA-4 antibodies could possibly induce the selective depletion of Tregs cells via Fc receptor-dependent cytotoxicity of macrophages or NK cells [117,118,119,120]. Although some reports have suggested that the depletion of Tregs is the primary reason for the antitumor response in CTLA-4 blockade [117,121], it is likely that both effector/memory T cell activation in SLOs and Treg depletion in tumor tissue cooperatively exert antitumor responses [122]. However, the tumoricidal effects of anti-PD-1 and anti-PD-L1 antibodies are likely mediated by the direct blocking of inhibitory PD-1 signals in effector T cells, and PD-L1 expressed on tumor cells or tumor-infiltrating immune cells, respectively [123,124,125].

Recently, the mechanism of action of these antibody-dependent therapies in SLOs, particularly in TDLNs, has been drawing increasing attention. The factors involved in T cell dynamics and T–DC interactions in LNs could thus be critical issues that determine the effects of ICBs.

## 8. Immune Checkpoint Inhibition and T Cell Motility

One possible mechanism by which ICBs restore antitumor immunity is by restoring the ability of antigen-dependent migration arrest in T cells, which could lead to stable contact with DCs and IS formation, followed by activation. Evidence from non-tumor models supports this hypothesis.

CTLA-4 modulates the motility of preactivated T cells both in vitro and in vivo [74]. The stimulation of CTLA-4 on activated T cells with agonistic antibodies has been shown to override the TCR-mediated stop signals on ICAM-1-coated plates [74]. The prevention of migration arrest was also observed in CTLA-4^+^ T cells on APCs expressing CD80/86 [75]. Potentiation of CTLA-4 by an agonistic anti-CTLA-4 antibody or ligation with CD80/86 can reverse the stop signal in CTLA-4^+^ preactivated T cells [69,74,75]. CTLA-4-mediated transient contact with APCs inhibits the formation of ZAP70 microclusters in T cells, leading to the suppression of IL-2 production and cell proliferation [75]. Conversely, antagonistic anti-CTLA-4 Fab fragments decreased the T cell motility [69] (Figure 2B). These results suggest that CTLA-4 transmits signals to suppress stop signals and, thus, continues T cell migration.

In a tolerance-inducing model of type I diabetes, blocking PD-1/PD-L1 and CTLA-4 leads to different consequences in terms of T cell motility [29]. Under tolerized conditions, islet antigen-specific T cells in pancreatic LNs sustained random motility without arrest, even in the presence of self-antigens [29]. The administration of anti-PD-1 or anti-PD-L1 antibodies abrogated tolerance by restoring the migration arrest of antigen-specific T cells [29] (Figure 2C). In contrast, anti-CTLA-4 antibodies had no significant influence on the motility of the tolerized T cells [29]. Although CTLA-4 plays an important role in inducing T cell tolerance [126], PD-1 is responsible for maintaining tolerance [127,128]. Therefore, CTLA-4 does not contribute to this process; thus, PD-1 blockade, and not CTLA-4 blockade, prevents the reversal of stop signals. This difference might be related to the fact that CTLA-4 inhibits signaling slightly downstream of TCR by recruiting PP2A, whereas PD-1 ligation directly inhibits the membrane-proximal events of TCR [111]. It has also been suggested that Tregs are involved in maintaining tolerance during CTLA-4 blockade [69].

Repeated T cell stimulation is known to induce the overexpression of immune checkpoint molecules, resulting in ‘immune exhaustion,’ which is associated with weakened immune functions [129]. In chronic viral infections and tumors, persistent antigen presentation and activation through the calcineurin/NFAT pathway induces the expression of the HMG-box transcription factor TOX in T cells [130,131,132,133,134]. A robust TOX expression reprograms CD8^+^ T cells transcriptionally and epigenetically to an exhausted state [130]. The epigenetic profile of exhausted T cells is clearly different from that of effector/memory T cells or anergic T cells. Importantly, ICBs transiently restores T cells from exhaustion, thereby promoting the clearance of viruses and tumors [135]. In a model of LCMV infection, PD-1 on exhausted T cells regulates cell motility, which is closely correlated with the maintenance of exhaustion in SLOs [126]. Chronic, but not acute, LCMV infection reduced the T cell motility in splenic white pulp. However, a subsequent PD-1 blockade restored the T cell motility, leading to viral reduction [136] (Figure 2D). PD-1/PD-L1 ligation recruits ICAM-1 to cSMAC, and promotes a stable IS formation in isolated T cells from this exhaustion model. These findings suggest that exhausted antiviral T cells enter a state of persistent PD-1 signaling through relatively prolonged contact with APCs, consequently reducing cytotoxicity [136]. The PD-1-mediated loss of motility in exhausted T cells is contrary to the persistent motility in tolerized T cells. The differences in CD8^+^ and CD4^+^ T cells, or the epigenetic profiles of exhausted versus tolerized/anergic T cells, might be a potential causative factor. Indeed, TOX and NR4A cooperatively enhance the expression of checkpoint molecules in exhausted T cells, whereas anergic T cells express NR4A without TOX [134,137]. Thus, the distinct regulation of checkpoint molecule expression seems to be one of the reasons for the differences in T cell motility induced by PD-1 blockade.

Together, the observation and evaluation of T cell dynamics in the context of ICBs are likely to provide valuable information for understanding the state of immunosuppression, tolerance, and exhaustion in various immunological settings. If this is the case, does checkpoint inhibition in tumors alter T cell dynamics as well?

## 9. CTLA-4 Inhibition and T Cell Dynamics in Tumors

Contrary to the initial assumption that CTLA-4 reverses the stop signal in antigen-dependent T cell arrest, some reports have shown that the administration of anti-CTLA-4 antibody increases T cell motility in LNs and tumors in the chronic phase [138,139]. Furthermore, anti-CD80 or anti-CD86 antibodies also increase T cell motility [138]. These results suggest that the blockade of CTLA-4 function inhibits the T cell adhesion to APCs in some situations. As CTLA-4 signaling induces the Rap-1-dependent clustering of LFA-1 and enhances T cell adhesion [140,141], CTLA-4 blockade may promote T cell motility by reducing stable adhesion.

Arriving in tumor tissues via the bloodstream, T cells infiltrate the interstitial space of the peri-tumoral region (which is rich in blood vessels, fibroblasts, immune cells, and extracellular matrices), followed by migration to the core region of the tumor (Figure 1C, D). In the parenchyma of the tumor core, T cells are arrested on tumor cells for killing them, and then further migrate to the neighboring live tumor cells to repeat the killing process [142,143]. There is conflicting evidence regarding whether the increased motility of CD8^+^ T cells by CTLA-4 blockade correlates with tumor suppression. In a melanoma model, CTLA-4 blockade increased CD8^+^ T cell motility within the tumor, which enhanced the brief contact with a large number of tumor cells, and the killing efficiency [138]. This finding supports the hypothesis that ICBs promote tumor rejection by enhancing T cell motility [144]. In contrast, in a breast tumor model, an increase in CD8^+^ T cell motility by CTLA-4 blockade did not promote tumor killing, whereas a combination with radiotherapy decreased CD8^+^ T cell motility and enhanced tumor injury [139]. The expression of ICAM-1 and NKG2D ligands on tumor cells was increased by this combination therapy, suggesting that these molecules could facilitate T cell arrest. It is considered that T cell cytotoxicity for the sufficient killing of tumor cells depends on the duration of physical contact [145]. Therefore, future studies are needed to examine the correlation between changes in T cell cytotoxicity and arrest/remigration in the tumor core region.

Within TDLNs, acute functional inhibition by the bolus administration of anti-CTLA-4 antibody increased the CD8^+^ T cell motility [138]. Similarly to the behavior of the exhausted T cells in the PD-1 blockade as described above [136], the alteration of T cell motility in SLOs potentially reflects the recovery of T cell function by CTLA-4 blockade. This finding is contrary to reports that show CTLA-4 blockade leading to T cell arrest [69]. However, it is difficult to compare these equally because of the difference in activation stage of T cells. Increased T cell motility may also contribute to antitumor activity by promoting the efficiency of the effector T cell egress from LNs, rather than antigen recognition. CTLA-4 signaling promotes integrin clustering and enhances the cell surface expression of CCR7 in T cells [146]. CTLA-4 blockade may, in turn, reduce CCR7 expression, which facilitates the T cell egress from LNs. In fact, this therapy promotes an increase in tumor-specific CD8^+^ T cells in tumors [147]. Therefore, the association between CTLA-4-targeted changes in T cell motility in LNs and antitumor effects must be further validated.

## 10. PD-1/PD-L1 Inhibition and T Cell Dynamics in Tumors

In a mouse melanoma model, live imaging of tumors after anti-PD-L1 antibody administration revealed the increased infiltration of adoptively transferred tumor-specific CD8^+^ T cells into the tumor core region with reduced motility and concordant elimination of tumor cells [148]. This finding indicates that PD-1/PD-L1 blockade promotes the migration of CD8^+^ T cells from the peri-tumoral region into the tumor core. Indeed, the presence of CD8^+^ T cells in the tumor core is closely correlated with the response rate of PD-L1 blockade in human patients [123]. Compared to anti-PD-L1 monotherapy, the combination of anti-PD-L1 and anti-CTLA-4 antibodies did not alter tumor eradication in the mouse melanoma model, despite an increase in highly motile T cells in the peri-tumoral area [148]. The addition of CTLA-4 blockade appeared to limit the enhancement of T cell migration from the peri-tumor region to the core. However, tumor-associated macrophages (TAMs) actively trap CD8^+^ T cells in the peri-tumoral area, which prevents migration into the tumor core [149]. Therefore, the combination of TAM depletion and PD-1 blockade could enhance infiltration, but decrease the motility of CD8^+^ T cells in the tumor core. Together, PD-1/PD-L1 regulates T cell dynamics both in the peripheral and core regions of tumors, and their blockade could lead to tumor eradication by modulating effector cell motility.

The modulation of cell motility is expected to synergize with adoptive immunotherapies, such as chimeric antigen receptor (CAR)-T cell therapy [123]. CAR-T cells express artificially constructed receptors that are specific to some tumor antigens, which provide strong signals and evoke cytotoxic activity upon binding to the target antigens. CAR-T cell therapy showed promising results in the treatment of hematologic malignancies. Although antitumor effects of CAR-T cells were also demonstrated in a variety of solid tumors, the efficacies are limited in clinical trials [150]. One of the reasons is thought to be due to the insufficient recruitment and infiltration of the transferred CAR-T cells into the tumor core region [151]. Trials are currently validating chemokine receptor-modified CAR-T cells to overcome this issue [152]. Together with this approach, the modulation of T cell dynamics using PD-1/PD-L1 blockade may be one of the solutions.

The PD-1/PD-L1 axis is thought to suppress antitumor responses via the direct interaction of PD-1 in antitumor CD8^+^ T cells with PD-L1 on tumor cells [153]. Therefore, the expression of PD-L1 in tumor tissues has been considered a promising marker for predicting the efficacy of immunotherapies using PD-1/PD-L1 blockade. However, there are many clinical cases in which anti-PD-1 and anti-PD-L1 antibodies fail to inhibit the growth of PD-L1-expressing tumors [154]. Previously, it was thought that the exhaustion of tumor-specific CD8^+^ T cells was one of the key factors that exacerbated the pathogenesis of tumors. However, it was clarified that an increase in stem-like CD8^+^ T cells, which are capable of self-renewal and differentiation with the TCF1^+^PD-1^+^ exhaustion phenotype in tumors, correlates with the efficacy of immunotherapies [155,156,157,158]. It is speculated that the stem-like CD8^+^ T cells are stored in some immunological niches of tumor tissues, which could contribute to sustainable antitumor responses, rather than transient responses [157,159]. These subsets are originally differentiated in TDLNs and enter the tumor from recirculation [160]. PD-1/PD-L1 blockade increases the precursor population in TDLN and expands them in the tumor [158,160,161]. In this process, interaction with dendritic cells is likely to promote the expansion of stem-like CD8^+^ T cells [160]. Presumably, blockade of PD-1-mediated inhibitory signals facilitates their proliferation.

Unfortunately, there is no direct evidence regarding the alteration of T cell motility in TDLNs in PD-1/PD-L1 blockade therapy. However, considering the close relationship between immune cell dynamics and ICBs, it is worth focusing on the motility changes of T cells in terms of the T–DC interaction within TDLNs, which may provide useful information to explain the efficacy of immunotherapy.

## 11. Conclusions

TDLNs have been shown to be a key site for inducing antitumor immunity wherein tumor-derived neoantigens are cross-presented by migratory DCs to antigen-specific CD8^+^ T cells. In recent years, the importance of TDLNs in tumor immunotherapy has become increasingly apparent, because the intensity of antitumor responses elicited by ICBs is largely dependent on TDLNs [162]. The TDLN resection completely abrogated the antitumor effect of PD-1/PD-L1 inhibition, indicating that TDLN plays a pivotal role in immune checkpoint therapy [163,164]. In addition, the intratumoral or subcutaneous administration of checkpoint inhibitors elicits more efficient responses in TDLNs, compared to those with systemic administration, providing therapeutically equivalent efficacy [165,166]. Based on these results, some studies have regarded TDLNs as the primary target for immunotherapy [164,166].

As discussed above, immune checkpoint molecules inhibit T cell activity, in part, by modulating the migration dynamics in LNs. In addition, ICBs induce an increase in the antitumor CD8^+^ T cell number in TDLNs, which enhances the antitumor response. We, thus, propose that ICBs optimize T cell motility and DC–T interaction in TDLNs, leading to the efficient activation or restoration of antitumor responses. The study of T cell dynamics in LNs is expected to be applied for elucidating the immune mechanisms in ICB therapy, and is beneficial for formulating the clinical treatment of tumors in combination with ICBs.

## Figures and Tables

**Figure 1 cancers-13-04616-f001:**
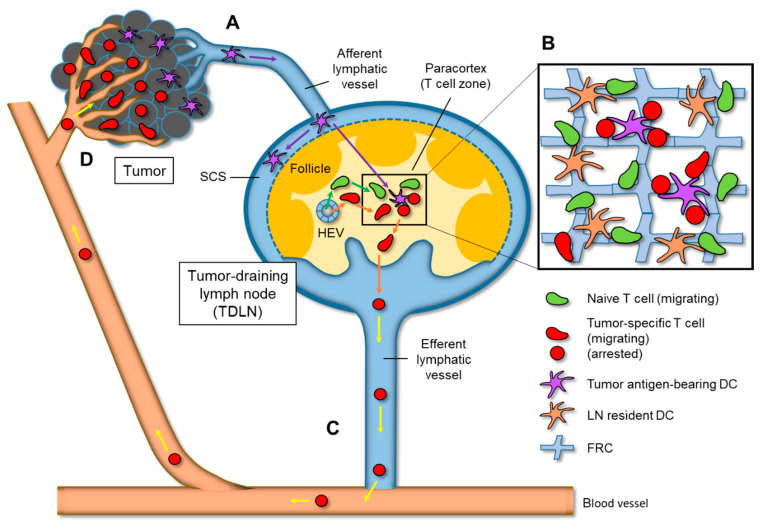
Immune cell trafficking in antitumor responses and motility changes in tumor-draining lymph nodes (TDLNs). (**A**) CCR7-expressing DCs that internalize tumor antigens in primary tumor migrate to TDLNs via afferent lymphatic vessels. Tumor antigen-bearing DCs reach the subcapsular sinus, followed by entry into the paracortical T zone. (**B**) Naive T cells that enter the paracortex via HEVs exhibit random migration in the interstitial space constructed by the fibroblastic reticular cell (FRC) meshwork and dendritic cells (DCs) when searching for antigens presented by DCs. When antigen-specific CD8^+^ T cell encounter CD103^+^ DCs (cDC1) displaying cognate antigens, they are arrested on the DC surface, forming an immunological synapse. After activation with sufficient TCR-dependent signaling, the motility of T cells is restored. (**C**) Activated/effector T cells or memory T cells leave the TDLN through the efferent lymphatic vessel and recirculate in the bloodstream. (**D**) Tumor antigen-specific CD8^+^ T cells transmigrate across tumor blood vessels into peri-tumoral region and infiltrate the tumor core region, where they kill tumor cells.

**Figure 2 cancers-13-04616-f002:**
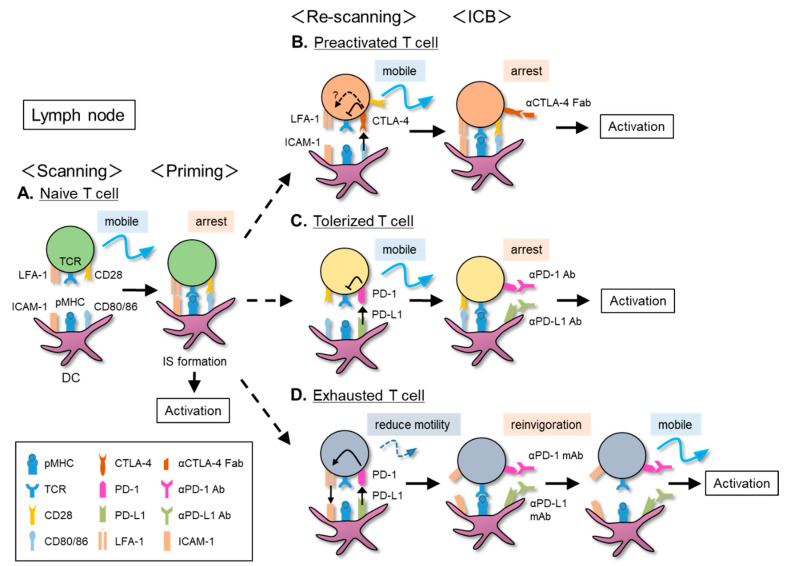
Schematic representation of the mobile/arrest-switching T cells during antigen recognition on dendritic cells (DCs), and the influence of immune checkpoint blockades (ICBs) in lymph nodes (LNs). (**A**) Naive T cells migrating in the paracortical T zone scan for antigen-presenting DCs. T cells that encounter cognate antigens are arrested on DCs and form the IS, composed of TCR–pMHC, costimulatory receptor-ligands, and adhesion molecules, which eventually activate T cells. (**B**) CTLA-4 expressed on preactivated T cells deprives CD28 of CD80/86-binding and inhibits the stop signal. Suppression of TCR-mediated stop signals weakens interaction with DC and retains T cell migration. Blockade of CTLA-4 ligation restores the ability of IS formation, leading to T cell activation. (**C**) In a tolerance-inducing type I diabetes model, PD-1/PD-L1 ligation maintains a tolerized and highly motile state in T cells by inhibiting the stop signal. PD-1/PD-L1 blockade promotes T–DC interaction restoring the ability of T cell arrest, which abolishes tolerance. As CTLA-4 is involved in the formation, but not the maintenance, of tolerance, CTLA-4 blockade does not alter T cell motility as well as tolerance. (**D**) In a model of chronic viral infection, persistent antigen exposure leads to T cell exhaustion. PD-1/PD-L1 ligation in exhausted T cells induces clustering of adhesion molecules, causing motility reduction. PD-1/PD-L1 blockade inhibits adhesion molecule clustering and restores T cell motility. PD-1/PD-L1 blockade also disrupts inhibitory signals, resulting in the reinvigoration of antiviral responses.

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
