# Peer review of "Motility Dynamics of T Cells in Tumor-Draining Lymph Nodes: A Rational Indicator of Antitumor Response and Immune Checkpoint Blockade"

_cancers, 2021, doi:10.3390/cancers13184616_

Round 1
Reviewer 1 Report
This review addresses the role of the immune checkpoint molecules (ICM) CTLA-4 and PD-1 in T cell migration and motility in regards to the induction and regulation of anti-tumor adaptive immune response in vivo. Particularly, an emphasis is made in the migration changes in tumor-draining lymph nodes (TDLN) during anti-tumor responses and the effect of immune checkpoint blockade (ICB) reagents used in anti-cancer therapies. This is a highly relevant topic in the context of ICB anti-tumor therapies that needs to be further clarified.
The review is well structured and the concepts are clearly exposed. These include the basic concepts of T cell migration during the development of adaptive immune responses, the role of different costimulatory and checkpoint molecules, and the particular case of anti-tumor immune responses. It includes two figures that help in the understanding of the above concepts.
The references are abundant, covering the topics addressed, and include both classic, early publications in the different topics, and recent relevant papers.
Comments:
There are two topics and several references that might be worth including in the review, as they can be relevant to the inhibitory mechanisms of CTLA-4 and PD-1:
One topic is that CTLA-4-mediated trogocytosis downregulates its ligands CD80 and CD86 in APC, impairing their activation ability [1,2].
The second topic is the fact that PD-L1 not only binds PD-1, but also can form dimers in cis with CD80 in the APC. The formation of PD-L1-CD80 dimers is relevant to CTLA-4 funtion, as interactions of CD80 with CD28 are spared, yet PD-L1/PD-1 and CD80/CTLA-4 trans-interactions and inhibitory signals are hampered [3,4].
- Ovcinnikovs, V.; Ross, E.M.; Petersone, L.; Edner, N.M.; Heuts, F.; Ntavli, E.; Kogimtzis, A.; Kennedy, A.; Wang, C.J.; Bennett, C.L.; et al. CTLA-4–mediated transendocytosis of costimulatory molecules primarily targets migratory dendritic cells. Sci. Immunol. 2019, 4, eaaw0902, doi:10.1126/sciimmunol.aaw0902.
- Qureshi, O.S.; Zheng, Y.; Nakamura, K.; Attridge, K.; Manzotti, C.; Schmidt, E.M.; Baker, J.; Jeffery, L.E.; Kaur, S.; Briggs, Z.; et al. Trans-endocytosis of CD80 and CD86: A molecular basis for the cell-extrinsic function of CTLA-4. Science 2011, 332, 600-603, doi:10.1126/science.1202947.
- Sugiura, D.; Maruhashi, T.; Okazaki, I.-m.; Shimizu, K.; Maeda, T.K.; Takemoto, T.; Okazaki, T. Restriction of PD-1 function by cis-PD-L1/CD80 interactions is required for optimal T cell responses. Science 2019, 364, 558-566, doi:10.1126/science.aav7062 %J Science.
- Zhao, Y.; Lee, C.K.; Lin, C.-H.; Gassen, R.B.; Xu, X.; Huang, Z.; Xiao, C.; Bonorino, C.; Lu, L.-F.; Bui, J.D.; et al. PD-L1:CD80 cis-heterodimer triggers the co-stimulatory receptor CD28 while repressing the inhibitory PD-1 and CTLA-4 pathways. Immunity 2019, 51, 1059-1073.e1059, doi:https://doi.org/10.1016/j.immuni.2019.11.003.
Author Response
Reviewer 1
Reply to Reviewer 1’s comments and relevant changes
We greatly appreciate the reviewer’s valuable comments to our manuscript.
- One topic is that CTLA-4-mediated trogocytosis downregulates its ligands CD80 and CD86 in APC, impairing their activation ability [1,2].
The description of CTLA-4–induced trogocytosis is added to section 6 (page 7, line 19-23).
- The second topic is the fact that PD-L1 not only binds PD-1, but also can form dimers in cis with CD80 in the APC. The formation of PD-L1-CD80 dimers is relevant to CTLA-4 funtion, as interactions of CD80 with CD28 are spared, yet PD-L1/PD-1 and CD80/CTLA-4 trans-interactions and inhibitory signals are hampered [3,4].
The description about cis-interaction between CD80 and PD-L1 is added to section 6 (page 7, line 41-48).
Reviewer 2 Report
The review by Kanda Y et al. describes the effects of the checkpoint therapy targeting CTLA-4 and PD-1 on T cell motility. The review is well written and a pleasant reading. It describes the important role of T cell motility in the interaction with antigen presenting cells and target cells in different settings of the immune response including the antitumor response, and it sheds light on the possible unconventional effect of checkpoint inhibitors on this aspect of T cell activity. I only suggest the following minor changes.
1) Since the authors focused on CTLA-4 and PD-1, but also briefly cite CD28, they might also briefly cite the possible role of ICOS (belonging to the same receptor family) in T cell motility and interaction with APC and target cells. For instance, it has been shown that ICOS triggering by its ligand (ICOSL) inhibits T cell motility. Moreover, ICOSL triggering by ICOS inhibits motility of dendritic cells and tumor cells. Intriguingly, ICOSL can bind also osteopontin, but this interaction stimulates migration of tumor cells (cite appropriate references).
2) Briefly describe cDC1 and cDC2 cells in section 2.
3) Briefly describe the role of the immunoreceptor-tyrosine-switch motif (ITSM) in PD-1 signaling in section 6
4) In section 8, the sentence “Administration of anti–PD-1 or anti–PD-L1 antibodies abrogated tolerance by restoring the migration arrest of antigen-specific T cells” appears to be contradicted by the successive sentence “thus, PD-1 blockade, but not CTLA-4 blockade, could reverse the stop signal” (in the same paragraph).
5) An abbreviation list might be helpful for readers.
Author Response
Reviewer 2
Reply to Reviewer 2’s comments and relevant changes
We thank the Reviewer for encouraging comments and valuable suggestions.
- Since the authors focused on CTLA-4 and PD-1, but also briefly cite CD28, they might also briefly cite the possible role of ICOS (belonging to the same receptor family) in T cell motility and interaction with APC and target cells. For instance, it has been shown that ICOS triggering by its ligand (ICOSL) inhibits T cell motility. Moreover, ICOSL triggering by ICOS inhibits motility of dendritic cells and tumor cells. Intriguingly, ICOSL can bind also osteopontin, but this interaction stimulates migration of tumor cells (cite appropriate references).
According to the reviewer's suggestion, we added the role of ICOS on T cell motility in section 5 (page 6, line 23-28).
- Briefly describe cDC1 and cDC2 cells in section 2.
We added a brief description of cDC1 and cDC2 in section 2 (page 4, line 6-7).
- Briefly describe the role of the immunoreceptor-tyrosine-switch motif (ITSM) in PD-1 signaling in section 6.
The role of ITSM in PD-1 signaling is described in Section 6 (page 7, line 35-37).
- In section 8, the sentence “Administration of anti–PD-1 or anti–PD-L1 antibodies abrogated tolerance by restoring the migration arrest of antigen-specific T cells” appears to be contradicted by the successive sentence “thus, PD-1 blockade, but not CTLA-4 blockade, could reverse the stop signal” (in the same paragraph).
Thank you for pointing out a serious error. There was an error in the last sentence of the section as you pointed out. We corrected it (page 8, line 47-48).
- An abbreviation list might be helpful for readers.
The list of abbreviations is added on page 11 line 35-49.
Reviewer 3 Report
In this manuscript, Kanda et al review the roles of Immune checkpoint blockade (ICB) in T cell motility and activation. Particularly, the authors focused on the importance of tumor-draining lymph nodes (TDLNs) in the anti-tumor responses via ICB therapy. The topic is certainly timely, with many important discoveries on this topic made in the past few years. Also, the topic should be of interest to a large number of scientists interested on the efficacy of ICI. The authors reviewed the recent literature well and summarized the key findings. Despite a couple of minor points (see below) this reviewer support publication of the manuscript once those remaining few points are solved:
-Some typos need check: page 7, “PL-L1” instead PD-L1.
-Some references are missing: page 8, “Evidence from non-tumor models supports this hypothesis.”
Author Response
Reviewer 3
Reply to Reviewer 3’s comments and relevant changes
We deeply appreciate your detailed review and comments regarding our paper.
- Some typos need check: page 7, “PL-L1” instead PD-L1.
The typos you pointed out are corrected (page 7, line 29). We also checked for spelling mistakes in the manuscript.
- Some references are missing: page 8, “Evidence from non-tumor models supports this hypothesis.”
Some references are added to section 8.
Reviewer 4 Report
The Review “Motility dynamics of T cells in tumor-draining lymph nodes: a rational indicator of antitumor response and immune check-point blockade” by Kanda et al.
The Manuscript by Kanda et al. describes a Review on the effects of tumor-draining lymph nodes on immune checkpoint blockade therapies. The Manuscript is logically organized, well written, and illustrated. It can be accepted in the present form, but as soon as adoptive immunotherapy shifts to application of the CAR-T cells or ex vivo expanded TILs, the addition of several paragraphs on it would significantly improve the Review.
Author Response
Reviewer 4
Reply to Reviewer 4’s comments and relevant changes
We thank the Reviewer for encouraging comments and valuable suggestion.
- The Manuscript by Kanda et al. describes a Review on the effects of tumor-draining lymph nodes on immune checkpoint blockade therapies. The Manuscript is logically organized, well written, and illustrated. It can be accepted in the present form, but as soon as adoptive immunotherapy shifts to application of the CAR-T cells or ex vivo expanded TILs, the addition of several paragraphs on it would significantly improve the Review.
As you pointed out, CAR-T cell therapy is expected to be applied to solid tumors in near future, so the mention of such a therapy is an important issue. We added the descriptions of CAR-T cell therapy in section 10 (page 10, line 35-45).